# Comparison of Fused Diffusion-Weighted Imaging Using Unenhanced MRI and Abbreviated Post-Contrast-Enhanced MRI in Patients with Breast Cancer

**DOI:** 10.3390/medicina59091563

**Published:** 2023-08-28

**Authors:** Kyeyoung Lee, Yeo Jin Jeong, Ki Seok Choo, Su Bong Nam, Hyun Yul Kim, Youn Joo Jung, Seung Ju Lee, Ji Hyeon Joo, Jin You Kim, Jin Joo Kim, Jee Yeon Kim, Mi Sook Yun, Kyung Jin Nam

**Affiliations:** 1Department of Radiology, Research Institute for Convergence of Biomedical Science and Technology, Pusan National University Yangsan Hospital, Pusan National University School of Medicine, Yangsan-si 50612, Republic of Korea; drkyeyoung@gmail.com (K.L.); kschoo0618@naver.com (K.S.C.); 2Department of Health Promotion Center, Pusan National University Yangsan Hospital, Yangsan-si 50612, Republic of Korea; jyj107@hanmail.net; 3Department of Plastic and Reconstructive Surgery, Research Institute for Convergence of Biomedical Science and Technology, Pusan National University Yangsan Hospital, Pusan National University School of Medicine, Yangsan-si 50612, Republic of Korea; subong71@hanmail.net; 4Department of Surgery, Research Institute for Convergence of Biomedical Science and Technology, Pusan National University Yangsan Hospital, Pusan National University School of Medicine, Yangsan-si 50612, Republic of Korea; isepa102@naver.com (H.Y.K.); gsjyj@hanmail.net (Y.J.J.); medicalarie@naver.com (S.J.L.); 5Department of Radiation Oncology, Research Institute for Convergence of Biomedical Science and Technology, Pusan National University Yangsan Hospital, Pusan National University School of Medicine, Yangsan-si 50612, Republic of Korea; hi_juji@daum.net; 6Department of Radiology, Research Institute for Convergence of Biomedical Science and Technology, Pusan National University Hospital, Pusan National University School of Medicine, Busan 49241, Republic of Korea; youdosa@naver.com (J.Y.K.); wwn35@hanmail.net (J.J.K.); 7Department of Pathology, Research Institute for Convergence of Biomedical Science and Technology, Pusan National University Yangsan Hospital, Pusan National University School of Medicine, Yangsan-si 50612, Republic of Korea; jeykim@pusan.ac.kr; 8Division of Biostatistics, Research Institute for Convergence of Biomedical Science and Technology, Pusan National University Yangsan Hospital, Yangsan-si 50612, Republic of Korea; msyun@pusan.ac.kr

**Keywords:** breast cancer, cancer detection, lesion conspicuity, diffusion-weighted imaging, abbreviated magnetic resonance imaging

## Abstract

*Background and Objectives*: To determine the percentage of breast cancers detectable by fused diffusion-weighted imaging (DWI) using unenhanced magnetic resonance imaging (MRI) and abbreviated post-contrast-enhanced MRI. *Materials and Methods*: Between October 2016 and October 2017, 194 consecutive women (mean age, 54.2 years; age range, 28–82 years) with newly diagnosed unilateral breast cancer, who underwent preoperative 3.0 T breast MRI with DWI, were evaluated. Both fused DWI and abbreviated MRI were independently reviewed by two radiologists for the detection of index cancer (which showed the most suspicious findings in both breasts), location, lesion conspicuity, lesion type, and lesion size. Moreover, the relationship between cancer detection and histopathological results of surgical specimens was evaluated. *Results*: Index cancer detection rates were comparable between fused DWI and abbreviated MRI (radiologist 1: 174/194 [89.7%] vs. 184/194 [94.8%], respectively, *p* = 0.057; radiologist 2: 174/194 [89.7%] vs. 183/194 [94.3%], respectively, *p* = 0.092). In both radiologists, abbreviated MRI showed a significantly higher lesion conspicuity than fused DWI (radiologist 1: 9.37 ± 2.24 vs. 8.78 ± 3.03, respectively, *p* < 0.001; radiologist 2: 9.16 ± 2.32 vs. 8.39 ± 2.93, respectively, *p* < 0.001). The κ value for the interobserver agreement of index cancer detection was 0.67 on fused DWI and 0.85 on abbreviated MRI. For lesion conspicuity, the intraclass correlation coefficients were 0.72 on fused DWI and 0.82 on abbreviated MRI. Among the histopathological factors, tumor invasiveness was associated with cancer detection on both fused DWI (*p* = 0.011) and abbreviated MRI (*p* = 0.004, radiologist 1), lymphovascular invasion on abbreviated MRI (*p* = 0.032, radiologist 1), and necrosis on fused DWI (*p* = 0.031, radiologist 2). *Conclusions*: Index cancer detection was comparable between fused DWI and abbreviated MRI, although abbreviated MRI showed a significantly better lesion conspicuity.

## 1. Introduction

Dynamic contrast-enhanced magnetic resonance imaging (DCE-MRI) is the most sensitive method for detecting breast cancer [1,2,3]. However, in the breast screening setting, access to DCE-MRI is relatively limited because of its prolonged inspection time, high cost, and long interpretation time. Thus, abbreviated MRI, which consists of a first post-contrast subtracted image and a maximum-intensity projection, is increasingly used in the clinical setting [4,5]. However, gadolinium-containing contrast agents cannot be administered for patients with renal dysfunction or previous adverse reactions to the contrast agents [6]. Many studies have documented the deposition and long-term retention of gadolinium in the deep nuclei of the brain, particularly after repeated exposure to gadolinium-based contrast agents [7,8,9]. For these reasons, non-contrast breast MRI with diffusion-weighted imaging (DWI) is currently used in research aiming to utilize it for screening purposes [10,11,12].

DWI, an unenhanced MRI technique, is one of the few non-invasive imaging modalities. It can evaluate microstructural data at the cellular level and enables the calculation of the apparent diffusion coefficient (ADC) associated with changes in tissues and intracellular structures [13,14]. DWI is useful for distinguishing between benign and malignant breast lesions [15,16]. A recent study has reported that DWI can potentially detect breast cancer and characterize breast lesions [17]. However, limitations in the clinical application of DWI, caused by breast anatomy, high susceptibility to artifacts, low spatial resolution, and spatial distortions, may result in decreased lesion conspicuity. [18]. These limitations can be overcome by fused high-b-value DWI and T1-weighted imaging (T1WI), which can provide both functional and anatomical information [19,20,21]. Shin et al. reported that fused high-b-value DWI and unenhanced T1WI could replace DCE-MRI with DWI as a screening tool [12]. Eghtedari et al. demonstrated that DWI with a b-value of 1000 s/mm^2^ showed better sensitivity for lesion detection [22]. Zhou, B. et al. reported that abbreviated MRI showed high sensitivity and specificity for the diagnosis of breast cancer, compared to full-scanning protocols [23]. However, few studies have compared the diagnostic performance of fused high-b-value DWI using unenhanced T1WI with that of abbreviated post-contrast-enhanced MRI [24,25]. Therefore, in the present study, we aimed to determine the percentage of breast cancers detectable by fused diffusion-weighted imaging (DWI) using unenhanced magnetic resonance imaging (MRI) and abbreviated post-contrast-enhanced MRI. 

## 2. Materials and Methods

### 2.1. Case Descriptions

Our institutional review board approved this retrospective study and waived the requirement for informed consent (IRB No. 05-2022-108). In a review of medical records between October 2016 and October 2017 at our institution, we identified 409 consecutive women with newly diagnosed unilateral breast cancer, who underwent preoperative breast MRI with DWI and subsequent breast cancer surgery. For patients who underwent multiple MRI examinations, only the first preoperative MRI was included. All MRI scans were reviewed by a senior radiologist (with 16 years of experience in breast imaging), who did not participate in this study. Moreover, 215 patients were excluded for the following reasons: underwent 1.5 T MRI (*n* = 74), underwent neoadjuvant chemotherapy (*n* = 54), underwent vacuum-assisted biopsy (*n* = 31), underwent excisional biopsy (*n* = 21), were lost to follow-up (*n* = 14), and had poor-quality MR images (*n* = 21). Finally, 194 unilateral breast cancers were identified in 194 patients, who constituted the study population (mean age, 54.2 years; range, 28–82 years) (Figure 1).

### 2.2. MRI Acquisition

MRI was performed using a 3.0-T system (Magnetom Skyra; Siemens Healthineers, Erlangen, Germany). A body radiofrequency coil was used for excitation, and a bilateral breast coil (16-channel or 18-channel phased-array coil) was used as the receiver. 

The MRI sequences included localizing, axial T2-weighted, axial diffusion-weighted, sagittal T1-weighted (before and five times after contrast medium injection), and axial T1-weighted (after dynamic enhancement) sequences. 

Dynamic MRI was performed using a three-dimensional, fat-suppressed, volumetric, interpolated breath-hold examination sequence with a parallel acquisition technique (generalized autocalibrating partial-parallel acquisition (GRAPPA) factor 4) before and five times after injecting a bolus of gadobutrol (0.1 mmol/kg, Gadovist; Bayer Schering Pharma, Berlin, Germany) at a rate of 2 mL/s and a subsequent 20 mL saline flush; all substances were administered using an automatic injector. Both breasts were examined in the sagittal plane on five dynamic images acquired every 90 s after the contrast medium injection. Additionally, subtraction images were obtained using pre- and post-contrast series to suppress the bright-fat signals. T1-volumetric interpolated breath-hold examination (vibe) images were acquired with repetition time (TR), 6.35 ms; echo time (TE), 2.92 ms; voxel size, 0.6 × 0.6 × 1.5 mm^3^; partition of slab, 224; flip angle, 24°; fat-suppression, quick fat sat technique; bandwidth, 330 Hz/pixel; plane, sagittal; and acquisition time, 90 s.

DWIs were acquired in the axial plane using the readout segmentation of long variable echo-trains (resolve) technique with b-values of 0, 1000 s/mm^2^; TR/TE, 7500/63 ms; voxel size, 2 × 2 × 4 mm^3^; number of acquisitions (NEX), 2; number of slices, 35; slice gap, 1.6 mm; number of readout segmentations, 7; diffusion mode, 3-scan trace; diffusion scheme, monopolar; echo spacing, 0.38 ms; bandwidth, 620 Hz/pixel; acquisition plane, axial; and acquisition time, 5 min 20 s. ADC maps were generated automatically using b-values by scanner. 

Pre-contrast T1-weighted sagittal images and axial diffusion-weighted images were transferred to a post-processing workstation (Syngovia, version of VB10, Siemens Healthineers) to generate sagittal fused images using a b-value of 1000 s/mm^2^.

### 2.3. Protocols of Fused High-b-Value DWI and Abbreviated MRI

The fused high-b-value DWI protocol consisted of sagittal fused images using a b-value of 1000 s/mm^2^ on the axial DWI and pre-contrast T1-weighted sagittal image, as well as axial, coronal, and sagittal DWI maximum-intensity projection (MIP) images.

The abbreviated MRI protocol consisted of T1-weighted sagittal images acquired once before and immediately after the contrast medium injection. These two image stacks were subtracted from first post-contrast subtracted images. Axial, coronal, and sagittal MIP images were reconstructed using the subtracted images.

### 2.4. Review Sessions

A senior radiologist identified the location of the index cancer on full dynamic contrast-enhanced MRI before the analysis. Two breast radiologists with 6 and 2 years of experience independently reviewed the fused high-b-value DWI and abbreviated MRI. During the review of the 194 image sets, the two radiologists were informed that all the patients had unilateral breast cancer. However, both radiologists were blinded to other clinical and pathological findings. Each radiologist evaluated two sets of MRI data, including fused high-b-value DWI and abbreviated MRI, according to the following criteria: detection of index cancer that showed the most suspicious finding in bilateral breasts, laterality (right vs. left), location (quadrant and subareolar area), lesion type on MRI (mass vs. non-mass enhancement), lesion size, and lesion conspicuity, using a 10-point scoring system. Scoring systems for lesion conspicuity were as follows: 2 (less than 25% of lesion borders definable), poorly delineable; 4 (25–50% of borders definable), moderately delineable; 6 (50–75% of borders definable), well delineable; 8 (more than 75% of lesion borders definable), excellently delineable; and 10 (100% of lesion borders definable), perfectly delineable. Cancer detectability scoring was defined as 0 or 1: 0, cancer was not marked; 1, cancer was correctly marked. Each radiologist reviewed the fused high-b-value DWI and the abbreviated MRI two weeks later to reduce the effect of the previous dataset review findings on subsequent review findings by the radiologist.

### 2.5. Pathologic Analysis

Histopathological information was acquired from pathology reports stored in the electronic archives of our institution. The final histopathological results of the surgical specimens were reviewed to determine the tumor type; invasiveness (invasive vs. in situ); levels of estrogen receptor (ER), progesterone receptor (PR), human epidermal growth factor receptor 2 (HER-2), and Ki-67; presence of lymphovascular invasion; presence of necrosis; presence of lymph node (LN) metastasis; and tumor size. Carcinoma in situ was defined as the proliferation of malignant epithelial cells that had not breached the myoepithelial layer. When malignant epithelial cells reached the basement membrane and invaded the adjacent stroma to a depth of 1 mm, microinvasion was deemed to be present. We classified the histopathological results of the surgical specimens into two groups based on invasiveness, and cancers exhibiting microinvasion were classified as invasive.

### 2.6. Statistical Analysis

The chi-square test was used to compare the detection rates of index cancers between fused high-b-value DWI and abbreviated MRI. A paired *t*-test was used to compare the lesion conspicuity between fused high-b-value DWI and abbreviated MRI. The chi-square test or Fisher’s exact test and the independent *t*-test were performed to ascertain the difference in cancer detectability between fused high-b-value DWI and abbreviated MRI based on the tumor type; invasiveness (invasive vs. in situ); levels of ER, PR, HER-2, and Ki-67; presence of lymphovascular invasion; presence of necrosis; presence of LN metastasis; tumor size; and lesion type on MRI. The κ statistic was used to determine the interobserver agreement on index cancer detection for fused high-b-value DWI and abbreviated MRI. Moreover, κ values of <0.20, 0.21–0.40, 0.41–0.60, 0.61–0.80, and >0.80 were considered to indicate poor, fair, moderate, good, and excellent agreement. In addition, the correlation coefficient was calculated to quantify the interobserver agreement for lesion conspicuity. According to Koo and Li, correlation coefficient values of <0.50, 0.50–0.74, 0.75–0.89, and >0.90 were indicative of poor, moderate, good, and excellent agreement, respectively. Statistical analysis was performed using SPSS version 28.0 (IBM Corp., Armonk, NY, USA), and a *p*-value < 0.05 was considered statistically significant.

## 3. Results

### 3.1. Breast Cancer Characteristics

The 194 study participants consisted of 161 patients with invasive ductal carcinoma, 15 with ductal carcinoma in situ (DCIS), 9 with mucinous carcinoma, 8 with invasive lobular carcinoma, and 1 with invasive apocrine carcinoma. The 179 invasive cancer sizes ranged from 0.1 to 12.5 cm, and the mean tumor size was 2.6 ± 1.62 cm. The 194 study participants comprised 169 (87.1%) masses and 25 (12.9%) non-mass enhancement. Among the mass lesions (*n* = 169), 131 (77.5%) masses were measured 2 cm or more, and 38 (22.5%) masses were less than 2 cm. Forty (20.6%) cases showed multifocality. 

### 3.2. Lesion Detection and Conspicuity

The detection rates of index cancers were higher on abbreviated MRI than on fused high-b-value DWI; however, there was no significant difference in index cancer detection between fused high-b-value DWI and abbreviated MRI (radiologist 1: 174/194 [89.7%] vs. 184/194 [94.8%], respectively, *p* = 0.057; radiologist 2: 174/194 [89.7%] vs. 183/194 [94.3%], respectively, *p* = 0.092) (Table 1) (Figure 2).

For fused high-b-value DWI, 14 (7.2%) index cancers were missed by both radiologists, 6 (3.1%) were missed by radiologist 1, 6 (3.1%) were missed by radiologist 2, and 168 (86.6%) were detected by both radiologists. For abbreviated MRI, 9 (4.6%) index cancers were missed by both radiologists, 1 (0.5%) was missed by radiologist 1, 2 (1.0%) were missed by radiologist 2, and 182 (93.8%) were detected by both radiologists. Both radiologists detected index cancers in five patients using abbreviated MRI; nevertheless, these cancers were not detectable on fused high-b-value DWI by both radiologists. These cancers showed a small size (≤1 cm), with persistent delayed enhancement on kinetic curve assessment (*n* = 3), non-mass enhancement with persistent delayed enhancement, and high-diffusion background signals (*n* = 2) (Figure 3) (Table 2).

Lesion conspicuity on abbreviated MRI was significantly higher than that on fused high-b-value DWI for both radiologists (radiologist 1: 9.37 ± 2.24 vs. 8.78 ± 3.03, respectively, *p* < 0.001; radiologist 2: 9.16 ± 2.32 vs. 8.39 ± 2.93, respectively, *p* < 0.001) (Table 1).

### 3.3. Interobserver Agreement

Both radiologists showed good (κ = 0.67, 95% confidence interval [CI]: 0.49–0.84) and excellent (κ = 0.85, 95% CI: 0.68–1.00) interobserver agreement based on index cancer detection on fused high-b-value DWI and abbreviated MRI, respectively. Regarding lesion conspicuity, the radiologists showed moderate (intraclass correlation coefficient [ICC] = 0.72, 95% CI: 0.64–0.78) and good (ICC = 0.82, 95% CI: 0.77–0.86) agreement on fused high-b-value DWI and abbreviated MRI, respectively.

### 3.4. Relations of Cancer Detection with Histopathological and Radiological Factors

The higher detection rate of index cancers was associated with tumor invasiveness on both fused high-b-value DWI and abbreviated MRI (*p* = 0.011 and *p* = 0.004, respectively; radiologist 1). For radiologist 2, tumor invasiveness (92.3%) was associated with the cancer detection rate on abbreviated MRI (*p* = 0.043), but not on fused high-b-value DWI (*p* = 0.190). Lymphovascular invasion on abbreviated MRI (32.0%, *p* = 0.032; radiologist 1) and necrosis on fused high-b-value DWI (37.1%, *p* = 0.031; radiologist 2) were associated with breast cancer detection. There were no significant differences in cancer detection based on the levels of ER, PR, HER-2, and Ki-67; presence of LN metastasis; tumor size; and lesion type on MRI (Table 3). 

Among the mass lesions, there were a total of 38 masses smaller than 2 cm, with radiologist 1 detecting 36 (94.7%) on abbreviated MRI and 33 (86.8%) on fused high-b-value DWI. Of the 131 masses that measured 2 cm or more, 127 (96.9%) masses were detected on abbreviated MRI, and 124 (94.7%) were detected on fused high-b-value DWI by radiologist 1. The detection rate of index cancers was not associated with tumor size on both abbreviated MRI and fused high-b-value DWI (*p* = 0.517 and *p* = 0.099, respectively).

## 4. Discussion

In this study, fused high-b-value DWI and abbreviated MRI showed comparable lesion detection rates, although fused high-b-value DWI showed a lower lesion conspicuity. 

DWI is an unenhanced MRI technique that has been used for the detection and characterization of breast cancer, requiring only two or three minutes [26]. Many occult breast cancers detected on DCE-MRI are also visible on DWI; therefore, it could be used as an alternative tool to avoid safety problems, to reduce the cost issues associated with gadolinium injection, and to avoid the longer scanning time of DCE-MRI [27]. For breast index cancer detection, there was no significant difference between fused high-b-value DWI and abbreviated MRI in this study. Likewise, in one recent study, the sensitivity/specificity of unenhanced abbreviated MRI based on DWI to detect breast cancers with less than 2 cm in diameter was comparable to that of post-contrast abbreviated MRI for breast cancer populations (reader 1: 89.9/97.6% and 95.5/90.6%, respectively; reader 2: 95.5/94.1% and 98.9/94.1%, respectively) [25]. Another recent study by Kim et al. reported unenhanced abbreviated MRI with DWI for detecting breast cancers was comparable to that of a post-contrast abbreviated MRI protocol. The sensitivity/specificity of post-contrast abbreviated MRI and qualitative and quantitative analyses of abbreviated DWI were 94.6%/94.2%, 84.8%/97.7%, and 87.0%/98.8%, respectively [28]. Based on these results, fused high-b-value DWI could be proposed as a promising detection tool for breast cancer.

Although fused high-b-value DWI showed comparable lesion detectability, its lesion conspicuity was lower than that of abbreviated MRI in our study. For DWI, the intravoxel partial volume of the unsuppressed fat signal causes measurement reduction, and can ultimately limit lesion conspicuity [29]. Chen et al. reported that the lesion conspicuity of DWI is not significantly influenced by varying the maximum b-value from 600 to 1000 s/mm^2^ (*p* = 0.303 and 0.840 for malignant and benign lesions, respectively) and is not different in conspicuity grades when compared among the three (600, 800, and 1000) b-values (*p* = 0.072) [30]. Likewise, we used a b-value of 1000 for fused DWI in this study, and the lesion conspicuity was lower than that of abbreviated MRI. For overcoming these limitations, computed higher-b-value images are needed in MRI protocols for breast cancer patients. One recent study revealed that the overall conspicuity of breast cancers on DWI is maximized using b-values ranging from 1200 s/mm^2^ to 1400 s/mm^2^ [31]. 

In our study, five cancers were detected using abbreviated MRI; however, these were not detected on fused high-b-value DWI by both radiologists. There were three mass lesions among these cancers, all of which showed a small size (≤1 cm). Kazama et al. reported a lower DWI sensitivity for the detection of cancers <1 cm (59%) than for the detection of cancers ≥1 cm (93%) [32]. This might be due to the partial volume effect of DWI itself. Better gradient and receiver coils could improve the spatial resolution and improve the sensitivity for small lesions. 

Invasive cancer showed a higher detection rate than carcinoma in situ on both fused high-b-value DWI and abbreviated MRI. More than half of DCIS (8/15, 53.3%) presented non-mass enhancement, which was affected by limited spatial resolution on DWI [33,34]. Furthermore, DCIS showed less angiogenesis and proliferation than invasive cancers, which could affect the uptake of contrast media [35]. Our study showed that lymphovascular invasion was associated with breast cancer detectability on abbreviated MRI. Lymphovascular invasion is a prognostic factor for poor clinical outcomes in patients with breast cancer [36,37]. Cheon et al. demonstrated that the adjacent vessel sign was more frequently found in patients with than without lymphovascular invasion [36]. In our study, prominent adjacent vessels were noted in many abbreviated MRIs, which were helpful for the detection of breast cancer. In addition, tumor necrosis was associated with the breast cancer detection rate on fused high-b-value DWI. According to Geschwind et al., DWI can be used to distinguish viable tumor cell zones from necrotic tumor zones. When tumor cells are viable, the cell membranes are intact and can restrict the diffusion of water molecules. Conversely, when tumor cells die, their membranes are broken and can no longer restrict the diffusion of water molecules. In this setting, water molecules are free to circulate within the tumor, and, thus, the signal intensity decreases in the necrotic portion of the tumor more than in the viable region [38]. The signal intensity difference between the necrotic and viable portions of the tumor might be helpful in detecting breast cancer on DWI.

This study has several limitations. First, it was a retrospective, single-center study conducted at a tertiary academic institution; thus, there might have been a selection bias. Second, this study was conducted in a cancer-only population, which might differ from the real-world setting. Although the radiologists were blinded to the DCE-MRI data, they knew that all participants were patients with breast cancer. This might have resulted in an overestimation of DWI performance. Third, all the patients had unilateral breast cancer. If the study included patients with bilateral breast cancer, the breast cancer detection rate might have been affected by the comparison with the other breast. Fourth, this study is focused on mainly index cancer detection. In the real-world setting, additional suspicious lesions are often detected through MR imaging, which can yield a different result of the diagnostic performance. Fifth, the *p*-values for cancer detection between fused high-b-value DWI using unenhanced MRI and abbreviated post-contrast-enhanced MRI were not statistically significant (*p* = 0.057 for radiologist 1, *p* = 0.092 for radiologist 2). Despite considering the possibility of increasing the cohort size, this study was limited by its predefined timeline from October 2016 to October 2017. Finally, all MRI examinations were performed after core needle biopsy, and, therefore, procedure-related changes, such as subacute hematoma, might have affected the cancer detectability on fused high-b-value DWI.

## 5. Conclusions

In conclusion, fused high-b-value DWI and abbreviated MRI showed a comparable index cancer detectability, although abbreviated MRI showed a significantly better lesion conspicuity, indicating the potential application of fused high-b-value DWI using unenhanced MRI as an alternative to abbreviated post-contrast-enhanced MRI.

## Figures and Tables

**Figure 1 medicina-59-01563-f001:**
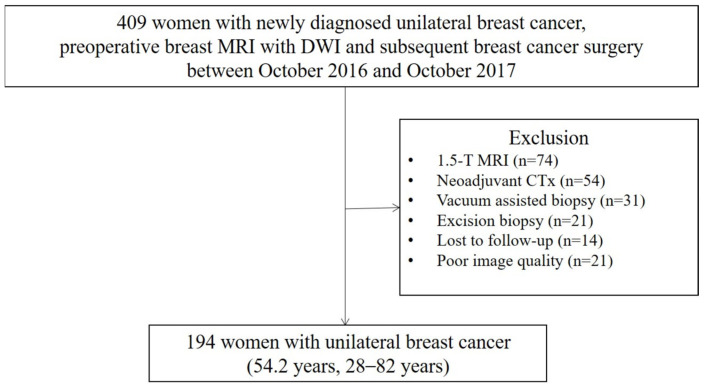
Flow chart of the patient selection. MRI = magnetic resonance imaging, DWI = diffusion–weighted imaging, CTx = chemotherapy.

**Figure 2 medicina-59-01563-f002:**
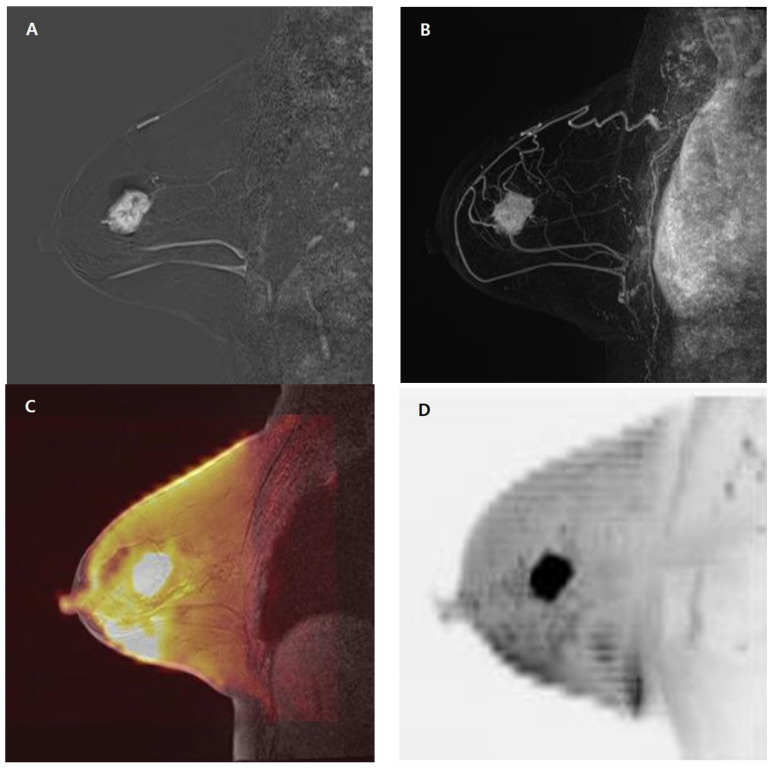
A 65-year-old woman with a 3.0 cm invasive ductal carcinoma in the right breast, ER/PR/HER-2 (−/+/−). (**A**–**D**). Sagittal subtracted image using the first post-contrast image (**A**) and sagittal maximum-intensity projection image (**B**) show a heterogeneous enhancing mass in the upper outer quadrant of the right breast. Sagittal fused diffusion-weighted image (**C**) and sagittal diffusion-weighted image with maximum-intensity projection (**D**) show an oval mass with focal diffusion restriction in the right breast. ER = estrogen receptor, PR = progesterone receptor, HER-2 = human epidermal growth factor receptor 2.

**Figure 3 medicina-59-01563-f003:**
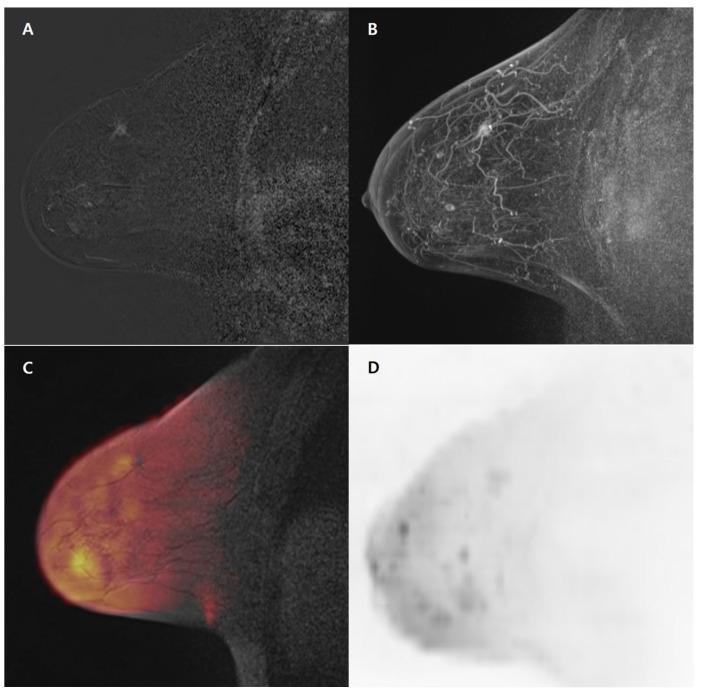
A 70-year-old woman with a 1.0 cm invasive ductal carcinoma in the right breast, ER/PR/HER-2 (+/+/−). (**A**–**D**). Sagittal subtracted image using the first post-contrast image (**A**) and sagittal maximum-intensity projection image (**B**) show irregular, spiculated, heterogeneously enhancing mass in the upper outer quadrant of the right breast. Sagittal fused diffusion-weighted image (**C**) and sagittal diffusion-weighted image with maximum-intensity projection (**D**) do not show any diffusion restriction in the right breast. ER = estrogen receptor, PR = progesterone receptor, HER-2 = human epidermal growth factor receptor 2.

**Table 1 medicina-59-01563-t001:** Index cancer detection and conspicuity on both fused high-b-value DWI using unenhanced MRI and abbreviated post-contrast-enhanced MRI in patients with unilateral breast cancer.

		Fused DWI	ABMR	*p*
Radiologist 1	Cancer detection (+),	174 (89.7)	184 (94.8)	0.057
	Cancer detection (−),	20 (10.3)	10 (5.2)	
	Conspicuity	8.78 ± 3.03	9.37 ± 2.24	<0.001
Radiologist 2	Cancer detection (+),	174 (89.7)	183 (94.3)	0.092
	Cancer detection (−),	20 (10.3)	11 (5.7)	
	Conspicuity	8.39 ± 2.93	9.16 ± 2.32	<0.001

**Table 2 medicina-59-01563-t002:** Radiologic characteristics of five unilateral breast cancers detected on abbreviated post-contrast-enhanced MRI, but not detected on fused high-b-value DWI by both radiologists.

	Lesion Type on MRI	Size on MRI (cm)	Kinetic Curve Assessment	High-Diffusion Background Signals	Background Parenchymal Enhancement
Patient 1	Mass	1.0	Persistent	(−)	Minimal
Patient 2	Mass	1.0	Persistent	(−)	Minimal
Patient 3	Mass	0.8	Persistent	(−)	Minimal
Patient 4	NME	5.5	Persistent	(+)	Minimal
Patient 5	NME	6.5	Persistent	(+)	Minimal

MRI = magnetic resonance imaging, DWI = diffusion-weighted imaging, NME = non-mass enhancement.

**Table 3 medicina-59-01563-t003:** Characteristics of 194 unilateral breast cancers and the relationship of breast cancer detection rates between fused high-b-value DWI and abbreviated MRI, based on histopathological and radiological factors.

Parameter	Number (%) of Participants (*n* = 194)	Radiologist 1		Radiologist 2	
		Fused DWI (+) (*n* = 174)	ABMR (+) (*n* = 184)	Fused DWI (+) (*n* = 174)	ABMR (+) (*n* = 183)
Invasiveness					
Invasive	179 (92.3)	164 (94.3)	173 (94.0)	162 (93.1)	171 (93.4)
In situ	15 (7.7)	10 (5.7)	11 (6.0)	12 (6.9)	12 (6.6)
*p*		0.011 ^2^	0.004 ^2^	0.190 ^2^	0.043 ^2^
ER					
Positive	149 (76.8)	131 (75.3)	141 (76.6)	131 (75.3)	140 (76.5)
Negative	45 (23.2)	43 (24.7)	43 (23.4)	43 (24.7)	43 (23.5)
*p*		0.171 ^2^	1.000 ^2^	0.171 ^2^	1.000 ^2^
PR					
Positive	127 (65.5)	112 (64.4)	120 (65.2)	112 (64.4)	120 (65.6)
Negative	67 (34.5)	62 (35.6)	64 (34.8)	62 (35.6)	63 (34.4)
*p*		0.344	1.000 ^2^	0.344	1.000 ^2^
HER-2					
Positive	39 (20.1)	37 (21.3)	38 (20.7)	37 (21.3)	38 (20.8)
Negative	155 (79.9)	137 (78.7)	146 (79.3)	137 (78.7)	145 (79.2)
*p*		0.376 ^2^	0.690 ^2^	0.376 ^2^	0.697 ^2^
Ki-67					
≥14%	108 (55.7)	100 (57.5)	102 (55.4)	98 (56.3)	101 (55.2)
<14%	86 (44.3)	74 (42.5)	82 (44.6)	76 (43.7)	82 (44.8)
*p*		0.136	1.000 ^2^	0.590	0.758 ^2^
Lymphovascular invasion					
Positive	62 (32.0)	59 (33.9)	62 (33.7)	58 (33.3)	61 (33.3)
Negative	132 (68.0)	115 (66.1)	122 (66.3)	116 (66.7)	122 (66.7)
*p*		0.086	0.032 ^2^	0.226	0.179 ^2^
Necrosis					
Positive	72 (37.1)	67 (38.5)	69 (37.5)	69 (39.7)	69 (37.7)
Negative	122 (62.9)	107 (61.5)	115 (62.5)	105 (60.3)	114 (62.3)
*p*		0.236	0.747 ^2^	0.031	0.749 ^2^
LN metastasis					
Positive	67 (34.5)	64 (36.8)	66 (35.9)	63 (36.2)	65 (35.5)
Negative	127 (65.5)	110 (63.2)	118 (64.1)	111 (63.8)	118 (64.5)
*p*		0.052	0.169 ^2^	0.149	0.336 ^2^
Tumor size (Invasive only) (cm)	2.6 ± 1.62 (range 0.1–12.5)	2.70 ± 1.62	2.68 ± 1.58	2.67 ± 1.59	2.72 ± 1.59
*p*		0.061 ^1^	0.588 ^1^	0.605 ^1^	0.127 ^1^
Lesion type on MRI					
Mass	169 (87.1)	152 (87.3)	162 (88.0)	154 (88.5)	161 (88.0)
Non-mass enhancement	25 (12.9)	22 (12.7)	22 (12.0)	20 (11.5)	22 (12.0)
*p*		0.765	0.097	0.087	0.142

Data are mean ± standard deviation or patient number (%). ^1^ Calculated using *t*-tests. ^2^ Calculated using Fisher’s exact tests. DWI = diffusion-weighted imaging, MRI = magnetic resonance imaging, ABMR = abbreviated magnetic resonance imaging, ER = estrogen receptor, PR = progesterone receptor, HER-2 = human epidermal growth factor receptor-2, LN = lymph node.

## Data Availability

The datasets generated or analyzed during this study are available from the corresponding author on reasonable request.

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
