# Peer review of "Comparison of Fused Diffusion-Weighted Imaging Using Unenhanced MRI and Abbreviated Post-Contrast-Enhanced MRI in Patients with Breast Cancer"

_medicina, 2023, doi:10.3390/medicina59091563_

Round 1
Reviewer 1 Report
This paper presents the findings of a study pertaining to the potential of unenhanced MRI as a breast cancer screening tool, however, there are a few major issues of concern that need to be addressed in my opinion.
First of all, while the fusion of DWI with another sequence may bring the benefit of improved spatial resolution, etc., in the context of screening this also has a negative side effect of decreased efficiency due to an overall increase in MRI scan time.
Secondly, some of your most important results, i.e. those concerning cancer detection rates, did not reach the level of statistical significance. To resolve this particular problem, I propose: 1) to either enlarge the patient cohort, or 2) to additionally present results for a specific subset of patients (e.g. invasive cancers >1 cm) where the calculated p-value could be <0.05.
Furthermore, the cited studies related to NSF and gadolinium chelates are obsolete for current clinical purposes, as the risk for this type of adverse reaction is negligible when marcocyclic chelates are used.
It is also unfortunate that the radiologists who evaluated the lesions were not blinded to the histopathological findings, thus precluding the measurement of parameters of diagnostic accuracy.
There are several minor linguistic errors within the manuscript that are fairly easy to correct, apart from one particularly confusing sentence in the "Introduction" section (l71-l74). I propose to rewrite it as follows:
However, limitations in the clinical application of DWI caused by breast anatomy, high susceptibility to artifacts, low spatial resolution, and spatial distortions, may result in decreased lesion conspicuity.
Author Response
Review Report Round 1
This paper presents the findings of a study pertaining to the potential of unenhanced MRI as a breast cancer screening tool, however, there are a few major issues of concern that need to be addressed in my opinion.
First of all, while the fusion of DWI with another sequence may bring the benefit of improved spatial resolution, etc., in the context of screening this also has a negative side effect of decreased efficiency due to an overall increase in MRI scan time.
- Thank you for your comment. In this study, we utilized a fused high b-value DWI protocol, which consisted of sagittal fused images using a b-value of 1000 s/mm². As you pointed out, an additional scanning time of 2 minutes were required to acquire this high b-value image. However, we believe that this time investment is justified if a fused high b-value DWI protocol can be used for screening without using a contrast agent, considering that dynamic contrast-enhanced magnetic resonance imaging takes an average of 20 to 30 minutes for the examination, resulting in a temporal advantage.
Secondly, some of your most important results, i.e. those concerning cancer detection rates, did not reach the level of statistical significance. To resolve this particular problem, I propose: 1) to either enlarge the patient cohort, or 2) to additionally present results for a specific subset of patients (e.g. invasive cancers >1 cm) where the calculated p-value could be <0.05.
- Thank you for your comment. As indicated, the p-values for cancer detection between fused high b-value DWI using unenhanced MRI and abbreviated post-contrast-enhanced MRI were 0.057 for Radiologist 1 and 0.092 for Radiologist 2. At the beginning of our study, we thought that this study would be meaningful if fused high b-vaule DWI using unenhanced MRI, which does not require the use of a contrast agent, showed a cancer detection rate similar to that of abbreviated post-contrast-enhanced MRI.
Despite considering the possibility of increasing cohort size, this study was limited by its predefined timeline from October 2016 to October 2017. Thus, the included patient population was maximized within this timeframe.
To mitigate the impact, we had attempted to focus solely on invasive cancers, excluding ductal carcinoma in situ. Additionally, among invasive cancers, we concentrated on those with a size of 1 cm or larger for statistical analysis. However, since a significant portion of the study population was already encompassed by invasive cancers larger than 1 cm, there was minimal variation in the overall sample size. As a result, the p-value did not reach significance. As you pointed out, I acknowledge that this limitation is inherent to the study.
Furthermore, the cited studies related to NSF and gadolinium chelates are obsolete for current clinical purposes, as the risk for this type of adverse reaction is negligible when marcocyclic chelates are used.
- Thank you for your comment. As you mentioned above, NSF is negligible when macrocyclic chelated are used. So the cited studies related to NSF and 7-9 references regarding NSF have been removed.
Macrocyclic gadolinium-based contrast agents (GBCA) such as gadoterate meglumine, gadobutrol, and gadoteridol have advantages over linear GBCAs. However, it has been reported that macrocyclic GBCAs are accumulating in the dentate nucleus, and although the difference is milder compared to the amount of accumulated contrast between linear and macrocyclic GBCAs, studies have also reported variations in the extent of accumulation among different macrocyclic GBCAs. Considering these side effects, it is believed that there may not be a complete alternative in this regard.
It is also unfortunate that the radiologists who evaluated the lesions were not blinded to the histopathological findings, thus precluding the measurement of parameters of diagnostic accuracy.
- I agree with your opinion. The design itself was intentionally crafted not only aim to assess the diagnostic performance of the images, but also focus solely on the detectability and conspicuity differences between two protocols: fused diffusion-weighted imaging (DWI) using unenhanced magnetic resonance imaging (MRI), and abbreviated post-contrast-enhanced MRI, when already aware of the presence of unilateral cancer. Therefore, the design was executed without being blinded to histopathological findings. And this limitation was described in the second limitation of our study in the “Discussion” section (Line 349-352).
Comments on the Quality of English Language
There are several minor linguistic errors within the manuscript that are fairly easy to correct, apart from one particularly confusing sentence in the "Introduction" section (l71-l74). I propose to rewrite it as follows:
However, limitations in the clinical application of DWI caused by breast anatomy, high susceptibility to artifacts, low spatial resolution, and spatial distortions, may result in decreased lesion conspicuity.
- Thank you for your comment. I have revised the sentence as you suggested.
Reviewer 2 Report
Comparison of Fused Diffusion-Weighted Imaging using Un- 2 enhanced MRI and Abbreviated Post-Contrast-Enhanced MRI 3 in patient with Breast Cancer
-In abstract: The results are nor well -presented in abstract, kindly revise and try to re-structure
-In introduction: “Dynamic contrast-enhanced magnetic resonance imaging (DCE-MRI) is the most sensitive method for detecting breast cancer [1-3]”: What about other sensitive methods?
“Recent studies have documented the deposition and retention of gadolinium in the deep nuclei of the brain,” It is not that recent if you refer to a reference published 6 years ago.
“Sabine et al. demonstrated that an abbreviated MRI protocol with high b-value DWI and DCE-MRI had comparable diagnostic accuracies in terms of lesion detection [25]. Ayami et al. reported that morphology and lesion extent showed high agreement between high resolution DWI and high resolution CE-MRI for malignant breast lesions [26]” Sabine and Ayami are not the proper references cited in the references list, You have to revise this and revise the whole citied sources.
- In methods you reviewed cases from 2016-2017 however we are in 2023, I wonder why did you choose this time period where new guidelines and new hospital’s strategies were implicated for breast cancer detection?
- Authors claim that the histopathological diagnosis was reviewed, however I realized that there is no any histopathogical authors in the authors list, also the criteria of revision and classification were not clearly mentioned, what WHO version used? What if this was any in discrepancy between the primary diagnosis and revised diagnosis?
- What about the previous radiological tools used for screening or early detection of your study participants?
Fine
Author Response
-In abstract: The results are nor well -presented in abstract, kindly revise and try to re-structure
à Thank you for your comment. As you pointed out, the results in the abstract were overly complex, so I have revised and summarized them for better clarity
-In introduction: “Dynamic contrast-enhanced magnetic resonance imaging (DCE-MRI) is the most sensitive method for detecting breast cancer [1-3]”: What about other sensitive methods?
à Thank you for your comment. In the evaluation of breast lesions, three modalities - mammography, ultrasonography, and magnetic resonance imaging (MRI) - are employed. Among these, MRI demonstrates an approximate 95% sensitivity for breast assessment, irrespective of breast density. In contrast, mammography exhibits an overall sensitivity of about 60%, which significantly decreases for dense breast parenchyma or small breast cancers. Ultrasonography can mitigate the limitations of mammography in certain patients. It has also been widely utilized for breast cancer screening and diagnosis, boasting a high sensitivity of 76%.
- Recent studies have documented the deposition and retention of gadolinium in the deep nuclei of the brain,” It is not that recent if you refer to a reference published 6 years ago.
à I agree with your opinion. As your suggestion, I have referred the references from articles published between 2017 and 2020. I have excluded the term "recent" and additionally revised scripts, incorporated the aspect highlighted in those studies, pertaining to “long-term retention”.
“Sabine et al. demonstrated that an abbreviated MRI protocol with high b-value DWI and DCE-MRI had comparable diagnostic accuracies in terms of lesion detection [25]. Ayami et al. reported that morphology and lesion extent showed high agreement between high resolution DWI and high resolution CE-MRI for malignant breast lesions [26]” Sabine and Ayami are not the proper references cited in the references list, You have to revise this and revise the whole citied sources.
à Thank you for your comment. I have removed previous references and added new ones that demonstrate higher sensitivity in cancer detection for not only high b-value DWI but also abbreviated MRI when compared to different b-value DWI or full scanning protocols, respectively, as you suggested.
- In methods you reviewed cases from 2016-2017 however we are in 2023, I wonder why did you choose this time period where new guidelines and new hospital’s strategies were implicated for breast cancer detection?
à Thank you for your comment. Back in 2016, there was a growing concern and research focus on the side effects of contrast agents. This led to a significant consideration of whether it would be possible to use MRI as a screening tool with high sensitivity without the need for contrast administration. Thus, a suitable protocol emerged involving diffusion-weighted imaging (DWI), which aimed to address the limitations of DWI by incorporating a high b-value fusion for MRI imaging. This experimental protocol was implemented for a year to assess its feasibility in our hospital. The study was initiated based on the data obtained during this temporary period. As you pointed out, we acquired the raw data during that specific timeframe. Subsequently, the processes of data curation and analysis followed, leading to a temporal gap between data selection and the commencement of the study.
- Authors claim that the histopathological diagnosis was reviewed, however I realized that there is no any histopathological authors in the authors list, also the criteria of revision and classification were not clearly mentioned, what WHO version used? What if this was any in discrepancy between the primary diagnosis and revised diagnosis?
à Thank you for your comment. The study was conducted at a tertiary academic institution, where a pathologist with 20 years of breast subspecialty experience evaluated the samples according to the WHO classification in effect at the time of the study. Histopathological findings were recorded in real-time in the electronic medical records, serving as the foundation for the progression of this study. As you suggested, the pathologist was included as a co-author.
- What about the previous radiological tools used for screening or early detection of your study participants?
à Thank you for your comment. In the realm of early detection of breast cancer, mammography and ultrasonography have been employed; however, as mentioned earlier (second comment), their sensitivities are not high. Hence, there is a movement towards utilizing MRI with its high sensitivity as a screening tool. However, MRI has been challenged by the use of contrast and lengthy scan times of 30 to 40 minutes. In order to ascertain whether MRI can overcome these limitations and serve as an effective screening tool, this study was conducted.
Round 2
Reviewer 1 Report
- Thank you for your comment. In this study, we utilized a fused high b-value DWI protocol, which consisted of sagittal fused images using a b-value of 1000 s/mm². As you pointed out, an additional scanning time of 2 minutes were required to acquire this high b-value image. However, we believe that this time investment is justified if a fused high b-value DWI protocol can be used for screening without using a contrast agent, considering that dynamic contrast-enhanced magnetic resonance imaging takes an average of 20 to 30 minutes for the examination, resulting in a temporal advantage.
I agree that a significant reduction of scan duration is to be seen when comparing fused high b-value DWI with a standard diagnostic DCE protocol. However, this advantage diminishes significantly in comparison to an abbreviated CE protocol, such as one that has been proposed by CK Kuhl, where MRI acquisition time was cca 3 minutes (see reference below).
Kuhl CK, Schrading S, Strobel K, Schild HH, Hilgers RD, Bieling HB. Abbreviated breast magnetic resonance imaging (MRI): first postcontrast subtracted images and maximum-intensity projection-a novel approach to breast cancer screening with MRI. J Clin Oncol. 2014 Aug 1;32(22):2304-10.
- Thank you for your comment. As indicated, the p-values for cancer detection between fused high b-value DWI using unenhanced MRI and abbreviated post-contrast-enhanced MRI were 0.057 for Radiologist 1 and 0.092 for Radiologist 2. At the beginning of our study, we thought that this study would be meaningful if fused high b-vaule DWI using unenhanced MRI, which does not require the use of a contrast agent, showed a cancer detection rate similar to that of abbreviated post-contrast-enhanced MRI. Despite considering the possibility of increasing cohort size, this study was limited by its predefined timeline from October 2016 to October 2017. Thus, the included patient population was maximized within this timeframe. To mitigate the impact, we had attempted to focus solely on invasive cancers, excluding ductal carcinoma in situ. Additionally, among invasive cancers, we concentrated on those with a size of 1 cm or larger for statistical analysis. However, since a significant portion of the study population was already encompassed by invasive cancers larger than 1 cm, there was minimal variation in the overall sample size. As a result, the p-value did not reach significance. As you pointed out, I acknowledge that this limitation is inherent to the study.
It is indeed unfortunate your study population could not have been increased so as to derive more statistically convincing results. On the other hand, as it remains the most problematic weakness of your study, this merits a mention in the "limitations" paragraph of the Discussion section.
I also have a comment pertaining to the last sentence of the first paragraph of the Introduction section (l64-65) where you claim that non-contrast breast MRI with diffusion-weighted imaging (DWI) is widely used to detect breast cancer.
The above-mentioned claim is followed by citation of three papers that only published results from specifically conducted research studies, not from breast MRI screening programs of any kind. The sentence in question should be changed to better reflect the current reality of breast MRI screening protocols as promising, but as of yet hypothetical procedures in terms of standard clinical practice.
Several linguistic errors still remain in the revised manuscript.
Author Response
Thank you for your comment. In this study, we utilized a fused high b-value DWI protocol, which consisted of sagittal fused images using a b-value of 1000 s/mm². As you pointed out, an additional scanning time of 2 minutes were required to acquire this high b-value image. However, we believe that this time investment is justified if a fused high b-value DWI protocol can be used for screening without using a contrast agent, considering that dynamic contrast-enhanced magnetic resonance imaging takes an average of 20 to 30 minutes for the examination, resulting in a temporal advantage.
I agree that a significant reduction of scan duration is to be seen when comparing fused high b-value DWI with a standard diagnostic DCE protocol. However, this advantage diminishes significantly in comparison to an abbreviated CE protocol, such as one that has been proposed by CK Kuhl, where MRI acquisition time was cca 3 minutes (see reference below).
Kuhl CK, Schrading S, Strobel K, Schild HH, Hilgers RD, Bieling HB. Abbreviated breast magnetic resonance imaging (MRI): first postcontrast subtracted images and maximum-intensity projection-a novel approach to breast cancer screening with MRI. J Clin Oncol. 2014 Aug 1;32(22):2304-10.
- I agree with your opinion. As mentioned before, abbreviated magnetic resonance imaging (MRI) is a solution to the long scan time drawback of dynamic contrast enhancement protocol, however, the disadvantage of using contrast agents is a limitation for utilizing this for routine screening purposes. Therefore, this study aimed to investigate whether contrast-agent-free approach like diffusion-weighted imaging (DWI), specifically fused high b-value DWI could be utilized for screening purposes. The objective of this study was to explore whether this approach could potentially serve as an alternative to abbreviated MRI.
- Thank you for your comment. As indicated, the p-values for cancer detection between fused high b-value DWI using unenhanced MRI and abbreviated post-contrast-enhanced MRI were 0.057 for Radiologist 1 and 0.092 for Radiologist 2. At the beginning of our study, we thought that this study would be meaningful if fused high b-vaule DWI using unenhanced MRI, which does not require the use of a contrast agent, showed a cancer detection rate similar to that of abbreviated post-contrast-enhanced MRI. Despite considering the possibility of increasing cohort size, this study was limited by its predefined timeline from October 2016 to October 2017. Thus, the included patient population was maximized within this timeframe. To mitigate the impact, we had attempted to focus solely on invasive cancers, excluding ductal carcinoma in situ. Additionally, among invasive cancers, we concentrated on those with a size of 1 cm or larger for statistical analysis. However, since a significant portion of the study population was already encompassed by invasive cancers larger than 1 cm, there was minimal variation in the overall sample size. As a result, the p-value did not reach significance. As you pointed out, I acknowledge that this limitation is inherent to the study.
It is indeed unfortunate your study population could not have been increased so as to derive more statistically convincing results. On the other hand, as it remains the most problematic weakness of your study, this merits a mention in the "limitations" paragraph of the Discussion section.
- Thank you for your comment. As you mentioned, we have included the limitation that p-values for cancer detection are not statistically significant in the discussion section.
I also have a comment pertaining to the last sentence of the first paragraph of the Introduction section (l64-65) where you claim that non-contrast breast MRI with diffusion-weighted imaging (DWI) is widely used to detect breast cancer.
The above-mentioned claim is followed by citation of three papers that only published results from specifically conducted research studies, not from breast MRI screening programs of any kind. The sentence in question should be changed to better reflect the current reality of breast MRI screening protocols as promising, but as of yet hypothetical procedures in terms of standard clinical practice.
- I agree with your opinion. Rather than being routinely used for breast screening, non-contrast breast MRI including DWI is currently used in research aiming to utilize it for screening purposes. According to this, the manuscript has been adjusted.